# A Computer Program for Assessing Histoanatomical Morphometrics in Ultra-High-Frequency Ultrasound Images of the Bowel Wall in Children: Development and Inter-Observer Variability

**DOI:** 10.3390/diagnostics13172759

**Published:** 2023-08-25

**Authors:** Tobias Erlöv, Tebin Hawez, Christina Granéli, Maria Evertsson, Tomas Jansson, Pernilla Stenström, Magnus Cinthio

**Affiliations:** 1Department of Biomedical Engineering, Faculty of Engineering, Lund University, 22363 Lund, Sweden; tobias.erlov@bme.lth.se (T.E.); magnus.cinthio@bme.lth.se (M.C.); 2Pediatrics, Department of Clinical Sciences, Faculty of Medicine, Lund University, 22185 Lund, Sweden; tebin.hawez@med.lu.se (T.H.); christina.graneli@med.lu.se (C.G.); maria.evertsson@med.lu.se (M.E.); 3Department of Pediatric Surgery, Skåne University Hospital Lund, 22185 Lund, Sweden; 4Department of Clinical Sciences Lund, Biomedical Engineering, Lund University, 22363 Lund, Sweden; tomas.jansson@med.lu.se; 5Clinical Engineering Skåne, Digitalisering IT/MT, RegionSkåne, 22185 Lund, Sweden

**Keywords:** bowel wall, computer program, histoanatomical morphometrics, ultra-high-frequency ultrasound

## Abstract

Ultra-high-frequency ultrasound (UHFUS) has a reported potential to differentiate between aganglionic and ganglionic bowel wall, referred to as histoanatomical differences. A good correlation between histoanatomy and UHFUS of the bowel wall has been proven. In order to perform more precise and objective histoanatomical morphometrics, the main research objective of this study was to develop a computer program for the assessment and automatic calculation of the histoanatomical morphometrics of the bowel wall in UHFUS images. A computer program for UHFUS diagnostics was developed and presented. A user interface was developed in close collaboration between pediatric surgeons and biomedical engineers, to enable interaction with UHFUS images. Images from ex vivo bowel wall samples of 23 children with recto-sigmoid Hirschsprung’s disease were inserted. The program calculated both thickness and amplitudes (image whiteness) within different histoanatomical bowel wall layers. Two observers assessed the images using the program and the inter-observer variability was evaluated. There was an excellent agreement between observers, with an intraclass correlation coefficient range of 0.970–0.998. Bland–Altman plots showed flat and narrow distributions. The mean differences ranged from 0.005 to 0.016 mm in thickness and 0 to 0.7 in amplitude units, corresponding to 1.1–3.6% and 0.0–0.8% from the overall mean. The computer program enables and ensures objective, accurate and time-efficient measurements of histoanatomical thicknesses and amplitudes in UHFUS images of the bowel wall. The program can potentially be used for several bowel wall conditions, accelerating research within UHFUS diagnostics.

## 1. Introduction

Ultrasonography is a diagnostic imaging method based on acoustic echoes and tissue-specific acoustic impedance. The amplitude, i.e., the strength of the reflected ultrasound waves, is described as hyperechoic with higher amplitudes (seen as white areas)—as in fat and collagen—hypoechoic, or anechoic (seen as gray or black areas, respectively) [1,2]. In daily medical use, ultrasound transducers transmitting 2–15 MHz are used, giving a good overview of the organs by imaging tissues to a depth of 2–20 cm. In contrast, UHFUS, with its much higher frequencies (30–50 MHz center frequency), allows for detailed imaging to depths of 0.1–5.0 mm [3,4]. UHFUS has been suggested to be promising for detailed diagnostics in several clinical areas [4,5,6]. Since the thickness of the bowel wall in small children is reported to be 0.3–2 mm, the use of UHFUS in the diagnosis of bowel diseases has also been suggested [7]. Hirschsprung’s disease (HD) is a congenital disease characterized by the absence of ganglia cells within two of the histoanatomical layers of the bowel wall. Currently, diagnosis of HD is by histopathological and immunohistological analyses of tissue biopsies [8], which are time-consuming and costly to perform. In order to replace the use of biopsies both during primary and surgical diagnosis with an instant and secure diagnostic method, the use of UHFUS is being explored [9]. Histoanatomical differences between aganglionic and ganglionic bowel wall have been suggested, and good correlations between histoanatomy and UHFUS of bowel wall specimens from HD patients have been confirmed [10]. Initial clinical observations have shown that UHFUS has the potential to delineate between aganglionic and ganglionic bowel wall [11]. The problem is that the assessment and calculation of histoanatomical measurements within UHFUS images could be both inaccurate and time-consuming, as well as associated with certain observer bias. To obtain reliable measurements of bowel wall layers quickly and easily, avoiding internal variability and limiting observer bias, a computerized assessment of the bowel wall within the UHFUS image is warranted. This would accelerate research by facilitating multiple measurements and enabling collection of amplitude information. Therefore, to be able to more precisely and objectively assess and calculate histoanatomical morphometrics, a computer program for UHFUS diagnostics is required. 

The main research objective of the study was to enable the assessment and automatic calculation of the histoanatomical morphometrics of the bowel wall in UHFUS images. The hypothesis was that a computer program would enable the assessment of relevant automatic calculations of the bowel wall’s histoanatomical morphometrics and that it would also deliver a high inter-observer correlation. 

The main aim of this study was to develop a computer program for the assessment and automatic calculation of the morphometrics of the bowel wall in UHFUS images. The secondary aim was to validate the computer program through inter-observer analyses between users. The main output variable was the ability of the computer program to measure the thicknesses and amplitudes of the bowel wall’s histoanatomical layers. The second output variable was the degree of inter-observer variability in the assessment of the thicknesses and amplitudes of the bowel wall. 

## 2. Materials and Methods

### 2.1. Settings

This was a developmental and validation study performed on UHFUS images of fresh ex vivo bowel wall specimens from children who underwent surgery for HD in a national referral pediatric surgery center for children with HD. The study was part of a larger translational project, involving pediatric surgeons, biomedical engineers and pathologists, and aiming to improve HD diagnostics by the use of UHFUS. 

### 2.2. Tissue Samples and Ultra-High-Frequency Ultrasound Images

All children with recto-sigmoid HD who underwent surgery with resection of the aganglionic bowel segment at a national Swedish referral center for HD, from April 2018 to December 2022, and for whom the guardians’ written consent had been obtained, were included in the study. A sample size of 20 was suggested by statisticians at the Department of Swedish Clinical Research Studies Forum South to be required, in accordance with guidelines for sample sizes in agreement analyses [12]. During the study period, a total of 37 children with HD underwent surgery. In accordance with a purposive and systematic sampling method, all these consecutive cases were evaluated for inclusion in the study. Ultimately, 23 of the 37 bowel wall specimens were included because they fulfilled the inclusion criteria by being rectosigmoid HD (<30 cm resected length), and because a successful imaging of both the aganglionic and ganglionic areas of the same specimen had been undertaken without technical obstacles. Of the 37 images, 14 were excluded because of aganglionosis extending more than 30 cm (*n* = 8), or due to an initial lack of standardized settings in the scanning UHFUS program (*n* = 6), leaving 23 images for analysis. 

The decision on surgical resection length was based on the pathologist’s analysis of intraoperatively taken fresh-frozen biopsies confirming the presence of ganglionic bowel wall. After surgical resection, the retrieved bowel wall segment was pinned to a cork mat and subjected to ex vivo UHFUS imaging from the serosal surface, at sites representing aganglionic and ganglionic bowel wall segments, respectively [11]. The specimen was then fixed in formalin and embedded in paraffin, and the presence of aganglionosis and ganglionosis was confirmed by histopathology (hematoxylin-eosin) and immunohistochemistry (S100 and calretinin) [9,11].

For ultrasound imaging, the Vevo^®^ MD ultrasound scanner, together with the UHF 70 transducer (both FUJIFILM VisualSonics Inc., Toronto, ON, Canada), delivering a center frequency of 50 MHz (bandwidth 29–71 MHz), was used.

Thus, for each patient, UHFUS images of both aganglionic and ganglionic bowel wall were available in the image database. These were exported for one-by-one measurements in the computer program for computerized assessments.

### 2.3. Computerized Program for Histoanatomical Morphometrics in the Bowel Wall

The computer program, consisting of a user interface in which the user can interact with the ultrasound images, was developed using MATLAB^®^ version 9.13 (R2022b, The MathWorks Inc., Natick, MA, USA) and its tools for user interfaces and functions, respectively, by researchers at the Department of Biomedical Engineering, Lund University. The purpose of the program was: (a) to enable multiple thickness measurements to be taken in a short amount of time; (b) to enable objective measurements of echo amplitudes (image whiteness) within different bowel wall layers in the ultrasound images; (c) to be easy to use by the pediatric surgeons on site; and (d) to be more time-efficient compared to manual measurements using calipers on the ultrasound scanner. 

To achieve these aims, the biomedical engineers and the pediatric surgeons worked in close collaboration, improving the user interface and program functionality continuously based on user experience and needs. In order to validate the software and ensure its security, requirements were defined and documented, outlining accomplishments and expected behavior of the program. After each update, the program’s new function was verified using simulated images which were assessed according to the specified requirements. In order to ensure that the code did not change during the study, a compiled version of the program was used. The software was developed within a research environment of an international leading research group for UHFUS with considerable experience in computer programming, especially in MATLAB. The engineers’ affiliation to one of Sweden’s largest research institutions for biomedical engineering secured a solid foundation of technical and experienced resources in computer programming. 

### 2.4. Measurements and Statistical Analysis

The thickness and amplitude of the layers in the bowel wall were measured in all included UHFUS images by two different observers using the computer program. In addition, ratios between measurements in different layers were calculated. The inter-observer variability of the computerized measurements was evaluated using the intraclass correlation coefficient (ICC) for two-way mixed effects, absolute agreement and multiple raters [13]. For testing the hypothesis, ICC values of less than 0.5 indicated poor reliability, values between 0.5 and 0.75 moderate reliability, values between 0.75 and 0.9 good reliability and values greater than 0.90 indicated excellent reliability [13]. In order to visualize agreement strength between observers, Bland–Altman plots were constructed [14]. These show the mean versus difference between observer measurements in each image. A mean level close to 0, and a narrow distribution within ± 2 SD, was considered to be a strong agreement, while a level with a mean that diverted considerably from 0, and with a wide distribution, was considered to be a poor agreement. Data management and statistical analyses were performed using Microsoft^®^ Excel 365 and MATLAB version 9.13 (R2022b, The MathWorks Inc., Natick, MA, USA). An appointed statistician from the government-supported Clinical Studies Sweden (https://kliniskastudier.se/english (accessed on 30 June 2023)) was consulted, and gave guidance on selecting statistical methods, as well as on the interpretation of results.

### 2.5. Ethical Considerations

Ethical approval was obtained from the local ethics review board (DNR 2017/769). Oral and written information was given, and the guardians’ written consent was obtained.

## 3. Results

### 3.1. Computerized Program

The end-product was a semi-automatic program delivering data in the form of mean and standard deviation (SD) on thickness and brightness amplitude (whiteness) of the muscularis externa and muscularis interna, but only amplitude for the submucosa. The thickness of the submucosa was not measured because the inner line of the submucosa could not be determined accurately enough by the use of the UHFUS transducer (70 MHz) (FUJIFILM VisualSonics Inc., Toronto, ON, Canada) with a center frequency of 50 MHz. This was mainly the result of the strong attenuation of high-frequency ultrasound while transducing the bowel wall from the serosa, often rendering the mucosa too deep to image with sufficient quality. For amplitude measurements within the submucosa, the program used a standardized area, calculated from the inner border of the muscularis interna to a delineation at 0.18 mm depth in the underlying submucosa. The program’s deliveries after user assessments are shown in Table 1.

Figure 1 shows the user interface after loading a UHFUS image. A range of interest (ROI) of 5 mm was decided upon for image quality and representation of the bowel wall, including continuity in the bowel wall.

Within the ROI, the inner and outer borders of the muscularis externa and the muscularis interna were marked manually by clicking within the image. The position and spacing between markings were chosen by the user and could be edited easily. At the same time, the software automatically created a full delineation of each layer within the ROI, by interpolating between these markings, and then displaying the layers on screen, see Figure 2.

Once satisfied, the operator stopped adding/editing points and saved the results. The results were displayed on screen, as well as saved in a Microsoft Excel spreadsheet. The thickness of each layer was automatically measured vertically at all lateral positions within the ROI, resulting in a mean, median and SD, based on about 160 measurements per layer. In order to calculate echo amplitudes, all image pixels within the ROI were used, resulting in a mean, median and an SD based on about 2000–10,000 measurements per layer.

The time required to import images into the program, to map out the histoanatomical layers and to extract all computerized measurements from an image of a bowel segment, took on average 2 min. This is in comparison with manual measurements using the scanner’s built-in caliper function, where extraction of three thickness measurements takes approximately 15 min, including selecting, manually measuring the different points, and transferring data from the images to external sheets for manual calculations. In addition, the ultrasound scanner does not permit other manual measurements to be taken, such as echo amplitudes, which the computer program does, which is why more data from images could be extracted by the program.

### 3.2. Inter-Observer Variability Tests

The results obtained by the computer program for the two observers are summarized in Table 2 and Table 3. There was an excellent agreement between observers in all thickness and amplitude measurements with an ICC range of 0.970–0.998. This agreement was also seen in the Bland–Altman plots, with narrow distributions and mean differences ranging from 0.005 to 0.016 mm (1.1–3.6%) from the overall mean for the thicknesses and 0 to 0.7 mm (0.0–0.8%) from the overall mean for amplitudes. Figure 3 and Figure 4 show the Bland–Altman plots for the thicknesses and amplitudes, respectively. 

## 4. Discussion

The aim of this developmental study was to describe the construction and evaluation of a computer program for measuring the thickness and ultrasound echo amplitudes in layers of the bowel wall. The purpose of this semiautomatic program was to enable and ensure more objective, accurate and time-efficient on-site measurements. This was achieved by automatically obtaining a maximum number of thickness and amplitude measurements (from all available image columns/pixels) within the chosen ROI, while keeping the user interaction to a minimum. The results show excellent agreement between observers, suggesting that the results provided by the computer program are observer-invariant. The program was deemed to be easy to use, as well as time-efficient.

Furthermore, the computer program enables automatic measurement of ultrasound-derived, tissue-specific parameters based on amplitude information, which is not currently possible with manual measurements. In this study, the average amplitude of each layer was measured. Besides being tissue-specific, this parameter is dependent upon several other factors, including the choice of scanner, transducer and settings. Therefore, a ratio between amplitudes in different layers could be a more relevant parameter, compensating for many of these undesired dependencies. However, there are several other amplitude-based parameters that have been shown to be robust and valuable for tissue characterization, including kurtosis, skewness and Nakagami-m distribution [2,3], that could be incorporated into the program. Hence, the use of computerized measurements is considered to be an asset for future work within this project, both from a scientific and a practical point of view.

The time efficiency of the computer program was found to be an advantage. Although times may vary depending on patient/image and on user experience, the developed program shows clear advantages in terms of user-operation time and the number of measurements per unit of time, compared to multiple manual measurements using an ultrasound scanner. Furthermore, the ultrasound scanner does not allow other manual measurements to be taken, such as echo amplitudes, but the computer program does.

The use of UHFUS to investigate the structure of the bowel wall has previously shown promise with regards to the delineation between aganglionosis and ganglionosis in patients with HD [11]. Research into the use of UHFUS in primary diagnostics of the bowel wall is ongoing [10]. Insertion of images into computer programs, as in this study, will enable more data to be assessed in a safe setting compared to data collected manually. Certainly, within the software of the Vevo^®^ MD, manual measurements of histoanatomical layers are possible. However, such manual assessment is time consuming, and the number of measurements needed to correspond accurately to the computerized calculation has not been evaluated. To be able to include both assessments in one program—thickness and amplitude measurements—is considered to be a tremendous advantage compared to manual calculations. In addition, the programmed calculations save observer time. Each assessment, collecting all data required, took on average just 2 min. 

Despite several advantages of computer calculations, it should be noted that the quality of images is of the utmost concern and that the examinator’s competence constitutes the main determining factor for all measurements. This study only addressed the inter-observer variability, not any inter-examinator’s variability that might be associated with the ultrasound scanning itself. Although computerized measurements might reduce potential variations in measurements between ultrasound images acquired by two different examinators, examiner-dependent variations are difficult to correct by such means and would be better addressed by the use of other methods. A potential variation between examinators could be interesting to investigate in future studies. Saving images for computerized assessment also makes fully blinded testing feasible. Comparisons of agreement in assessments of aganglionic and ganglionic tissues, respectively, were not undertaken since all single measurements indicated high agreements. However, this could be of interest in future studies comparing UHFUS features of aganglionic and ganglionic bowel wall.

One strength of this study is the novelty of the technique, since no automatic assessment by a computer program for bowel wall assessment using UHFUS has been proposed previously. The program was developed in close collaboration between end users who are well-experienced clinicians operating daily on bowel walls, and biomedical engineers with substantial experience in constructing and implementing similar programs in other imaging applications. Several adjustments of the computerized program were made during the development process, following repeated discussions about combined clinical relevance and future research topics, with respect to detailed amplitude analyses and testing. Another strength of our study is the fact that the developed program could potentially be used not only for assessment of the bowel wall in children, but also for adults, and in several additional conditions, e.g., delineations of inflammatory bowel disease, and in endo-luminal UHFUS imaging.

A limitation of our study is that the computer program does not enable calculations of the whole bowel wall thickness to be made. This was because the UHF 70 transducer generally provided poor image quality at the depth of the mucosa. A program is under development which includes possibilities to also explore the full thickness of the bowel wall using a 30 MHz center frequency transducer (UHF 48, bandwidth 20–46 MHz, FUJIFILM VisualSonics Inc., Toronto, ON, Canada). For interested UHFUS users, the plan is to make the program for automatic assessment of UHFUS images available in future (after validation of different center frequencies) within a complete user package, including instructions.

## 5. Conclusions

This computer program enables and ensures more objective, accurate and time-efficient on-site measurements of the histoanatomical layers of the bowel wall. It also provides unique possibilities to quantify amplitudes in different bowel wall layers assessed by UHFUS imaging. The inter-variability analyses show excellent agreement between observers, suggesting that the results provided by the program are observer-invariant. The computer program was deemed to be easy to use, as well as time-efficient.

## Figures and Tables

**Figure 1 diagnostics-13-02759-f001:**
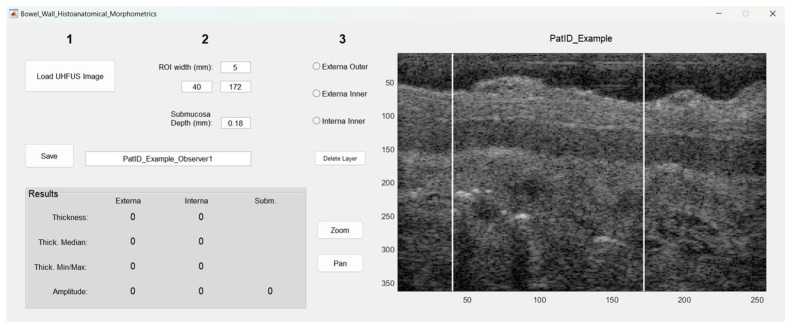
The user interface of the developed program. After loading an image and choosing a suitable range of interest, the user delineates the outer and inner border of the muscularis externa, and the inner border of the muscularis interna, by clicking within the image. The results are shown automatically in parallel to the delineation. Once finished, the user saves the results, and then repeats the process for the next image. Numbers 1–3 in the top refer to the order of how the user is intended to use the program.

**Figure 2 diagnostics-13-02759-f002:**
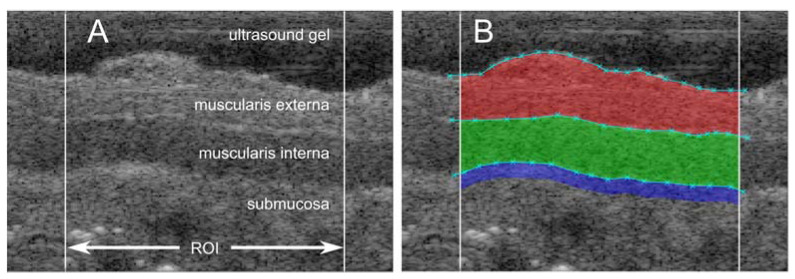
(**A**) Ultrasound image (zoomed) where different layers of the bowel wall are shown, together with the 5 mm wide range of interest. (**B**) The same ultrasound image with user markings (cyan crosses) and the resulting areas of the muscularis externa (red), muscularis interna (green) and submucosa (blue) with a predefined analysis depth of 0.18 mm (blue).

**Figure 3 diagnostics-13-02759-f003:**
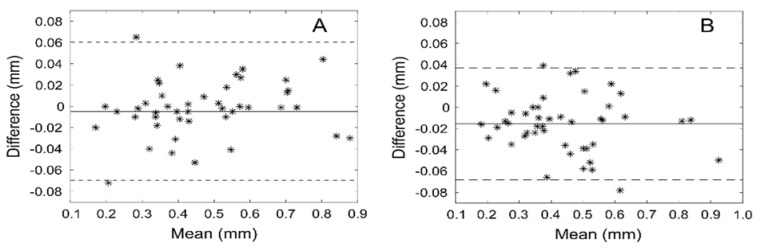
Bland-Altman plots that show the differences between users’ measured thicknesses against their mean for muscularis externa (**A**) and muscularis interna (**B**) mm—millimeters.

**Figure 4 diagnostics-13-02759-f004:**
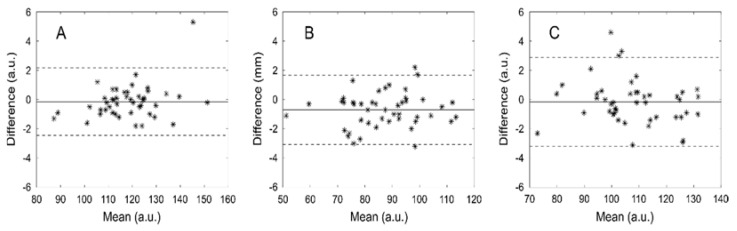
Bland-Altman plots that show the differences between users’ measured amplitudes against their mean for muscularis externa (**A**), muscularis interna (**B**) and submucosa (**C**) a.u.—albitrary units.

**Table 1 diagnostics-13-02759-t001:** The program’s deliveries after user assessments.

	Thickness	Amplitude
Definition	Number of vertical pixels multiplied with the length of one pixel (in mm)	The unitless value of a pixel representing ‘whiteness’
Measurements	Mean, Median, SD (Number of measurements per layer was about 160)	Mean, Median, SD (Number of measurements per layer was about 2000–10,000)
Layers included in analysis	Muscularis externa Muscularis interna	Muscularis externa Muscularis interna Submucosa (first 0.18 mm)

**Table 2 diagnostics-13-02759-t002:** The resulting thickness for the two muscle layers and the ratio between them. Presented values are the overall mean and standard deviation (SD); the difference between observers; and the intraclass correlation coefficient (ICC).

	Overall Mean (SD) (mm)	Difference, Mean (SD) (mm)	ICC
Muscularis externa	0.477 (0.185)	0.005 (0.033)	0.992
Muscularis interna	0.439 (0.166)	−0.016 (0.026)	0.992
Ratio (interna/externa)	1.031 (0.479)	−0.014 (0.167)	0.970

**Table 3 diagnostics-13-02759-t003:** The resulting amplitudes for the two muscle layers and the ratio between them. Presented values are the overall mean and standard deviation (SD); the difference between observers; and the intraclass correlation coefficient (ICC). Muscularis interna (int), Muscularis externa (ext), submucosa (sub). a.u.—arbitrary units.

	Overall Mean (SD) (a.u.)	Difference Mean (SD) (a.u.)	ICC
Muscularis externa	118.2 (12.5)	−0.1 (1.1)	0.998
Muscularis interna	87.2 (13.4)	−0.7 (1.2)	0.997
Submucosa	107.7 (14.0)	−0.2 (1.5)	0.997
Ratio int/ext.	0.738 (0.090)	−0.005 (0.011)	0.995
Ratio sub./ext.	0.917 (0.124)	0.000 (0.015)	0.996
Ratio int./sub.	0.810 (0.082)	−0.006 (0.017)	0.988

## Data Availability

The data that support the findings of this study are available from the corresponding author (P.S.) upon request.

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
