# Peer review of "A Computer Program for Assessing Histoanatomical Morphometrics in Ultra-High-Frequency Ultrasound Images of the Bowel Wall in Children: Development and Inter-Observer Variability"

_diagnostics, 2023, doi:10.3390/diagnostics13172759_

Round 1

Reviewer 1 Report

• The "Introduction" part of the study should be expanded, considering the research objectives, problems, and hypotheses. • The primary output/endpoint variable(s)/measurement(s) of the study should be defined.  • How was the sample size determined? This information should be explained in the Materials and Methods section.  • Which sampling (probable or non-probable, etc.) method was used in the study?  • Statistical tests for hypothesis testing and their assumptions should be specified in the study's statistical analysis in the Materials and Methods section.  • More detailed information about the environment in which the software will be developed should be presented. • How and/or which techniques will be used to ensure the security of the proposed software should be specified. • The details (version, license number, etc.) of the statistical package(s) or program(s) should be given in the section of "Data Analysis or Statistical Analysis". • The "Introduction" part of the study should be expanded, considering the research objectives, problems, and hypotheses. • The primary output/endpoint variable(s)/measurement(s) of the study should be defined.  • How was the sample size determined? This information should be explained in the Materials and Methods section.  • Which sampling (probable or non-probable, etc.) method was used in the study?  • Statistical tests for hypothesis testing and their assumptions should be specified in the study's statistical analysis in the Materials and Methods section.  • More detailed information about the environment in which the software will be developed should be presented. • How and/or which techniques will be used to ensure the security of the proposed software should be specified. • The details (version, license number, etc.) of the statistical package(s) or program(s) should be given in the section of "Data Analysis or Statistical Analysis".

Author Response

Dear Reviewer 1

We are very grateful for your review and the opportunity to improve the manuscript accordingly. The requested information is added and the highlighted issues clarified. Please see the attached table for a structured overview of our response. All changes are colored.

Best regards

Pernilla Stenström

Reviewer 2 Report

The aim was to describe the development of a computer program, little is mentioned on this development e.g. this article does not make the reader able to redo the programming work.  The aim needs to be refrased.  The article is on assessing and maybe fine tuning of a computer program.    It would also be of interest to read on how the two different bowl sets(with and without ganglia ) performed in the computerized measurements

Author Response

Dear Reviewer 2

We are grateful for your review and the opportunity to improve the manuscript. We have made changes according to your suggestions. Please see the attached table for a structured response and overview of all changes, which are color marked in text.

Best regards

Pernilla Stenström

Round 2

Reviewer 1 Report

Accept in present form